# A Unified View on Neural Message Passing with Opinion Dynamics for Social Networks

## Abstract

Social networks represent a common form of interconnected data frequently depicted as graphs within the domain of deep learning-based inference. These communities inherently form dynamic systems, achieving stability through continuous internal communications and opinion exchanges among social actors along their social ties. In contrast, neural message passing in deep learning provides a clear and intuitive mathematical framework for understanding information propagation and aggregation among connected nodes in graphs. Node representations are dynamically updated by considering both the connectivity and status of neighboring nodes. This research harmonizes concepts from sociometry and neural message passing to analyze and infer the behavior of dynamic systems. Drawing inspiration from opinion dynamics in sociology, we propose ODNET, a novel message passing scheme incorporating bounded confidence, to refine the influence weight of local nodes for message propagation. We adjust the similarity cutoffs of bounded confidence and influence weights of ODNET and define opinion exchange rules that align with the characteristics of social network graphs. We show that ODNET enhances prediction performance across various graph types and alleviates oversmoothing issues. Furthermore, our approach surpasses conventional baselines in graph representation learning and proves its practical significance in analyzing real-world co-occurrence networks of metabolic genes. Remarkably, our method simplifies complex social network graphs solely by leveraging knowledge of interaction frequencies among entities within the system. It accurately identifies internal communities and the roles of genes in different metabolic pathways, including opinion leaders, bridge communicators, and isolators.

## 1 Introduction

Sociometry is a quantitative method used in social psychology and sociology to describe social relations (Moreno, 1934; 2012). In his pioneering work, Moreno (1934) conceptualized a graph as an abstract representation of a group's structure. The term *social network* was later coined to describe a system comprising individual social actors and the social ties among them (Proskurnikov & Tempo, 2017). The development of cybernetics has led to increased attention to the study of messages and communication within society (Wiener, 1988). Statistical physics has contributed by introducing methods and tools from dynamical systems theory, giving rise to the field of *sociodynamics* (Weidlich, 2006; Helbing, 2010).

Graph neural networks (GNNs), on the other hand, are rooted in the same basic structure as social networks: graphs. The primary challenge in designing GNN models lies in effectively aggregating information based on local interactions for efficiently extracting hidden representations. This design philosophy has been generalized as *neural message passing* (MP; Gilmer et al. (2017)) and later became a fundamental feature extraction unit of graph-structured data for aggregating features of neighbors during network propagation.

This work explores the connection between these two fields by delving into *opinion dynamics*, a subfield of sociodynamics. We establish a link between the French-DeGroot (FD) model (French Jr, 1956; DeGroot, 1974) and MP, emphasizing a shared phenomenon that in both FD models and MPs a network converges exponentially to a stable state when it exhibits strong local connectivities, a property that is frequently observed in hypergraphs. Moreover, we draw inspiration from the

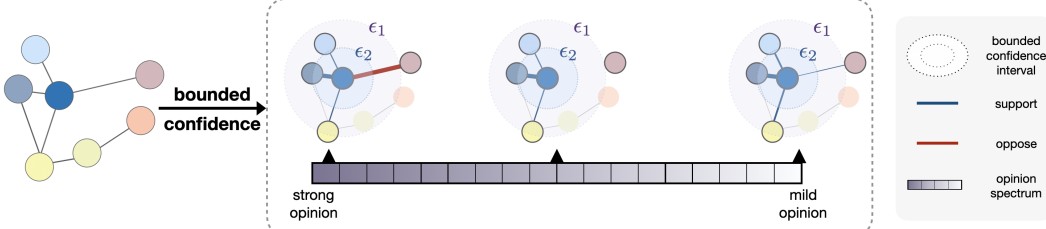

Figure 1: Given a graph with initial connections, ODNET is defined with bounded confidence to update the influence weights by the graph's position on the opinion spectrum.

Hegselmann-Krause (HK) model (Rainer & Krause, 2002) and incorporate the concept of *bounded confidence* into our novel MP formulation termed ODNET, which features a confidence filtration mechanism on initial edge connections. Based on the similarity of node pairs and connection proximity, ODNET aggregates neighboring information through automatic adjustment on edge weights. With piecewise MP schemes, the model strengthens, weakens, or removes initial links within the graph. Additionally, it allows for assigning negative weights to capture adverse perspectives from neighboring nodes with substantial disparities. This feature is essential when investigating heterophilic networks, where connected entities exhibit dissimilar characteristics.

The opinion dynamics-inspired propagation scheme can take on a static or dynamic nature, contingent on the choice of the similarity measure applied to nodes. The new mechanism emulates the dynamic spreading of opinions in a community, where effective communications converge individuals or agents towards consensus or several predominant viewpoints. For instance, in microbial communities, analyzing the co-occurrence network of metabolic genes across various species can reveal the potential biologically key genes act as predominant roles (Liu et al., 2018). When exchanging ideas, individuals tend to support opinions akin to their own. Depending on the position of the community on the opinion spectrum (Figure 1), when encountering significantly divergent thoughts, individuals may choose to disregard or oppose them. For instance, researchers typically focus on studies within their expertise but would love to learn new perspectives from other domains, whereas politicians usually have strong conflicts and resist propositions from competing parties.

The proposed approach offers an enhanced description of neighborhood influence across various scenarios by categorizing the relationships among nodes into three levels. We assess the versatile ODNET across three categories of graphs: homophilic graphs, heterophilic graphs, and hypergraphs, each characterized by its unique properties. Empirically, the introduced piecewise aggregation behavior enhances the performance of previously established MP methods (such as GCN (Kipf & Welling, 2017), GAT (Veličković et al., 2018), and HGNN (Gao et al., 2022)). We also demonstrate our model's capacity to progressively simplify graph structures, a crucial feature for deciphering complex social networks. To underscore the practical significance of this simplification capability, we provide a concrete application in the field of microbiology. Our model is shown to effectively prune weak connections among genes, thus extracting biologically relevant genes and connections from co-occurrence networks of metabolic genes.

## 2    NEURAL MESSAGE PASSING FOR GRAPHS AND HYPERGRAPHS

**Graph**    A graph $\mathcal{G}[\boldsymbol{W}] = (\mathcal{V}, \mathcal{E}[\boldsymbol{W}])$ of $N$ nodes can be associated with any square non-negative weight matrix $\boldsymbol{W} \in \mathbb{R}^{N \times N}$, where $\mathcal{V}$ represents the node set and $\mathcal{E}$ is the edge set. An edge $(v_i, v_j) \in \mathcal{E}$ if and only if $w_{ij} > 0$. We denote $x_i$ as the feature of node $v_i$ or the opinion of individual $i$. $\mathcal{G}$ is *strongly connected* (Godsil & Royle, 2001) if there exists a path from every node to every other node. A cycle is a directed path that both begins and ends at the same node with no repeated nodes except for the initial/final one. The length of a cycle is defined by the number of edges in the cyclic path. The periodicity of a graph is defined as the smallest integer $k$ that divides the length of every cycle in the graph. When $k = 1$, $\mathcal{G}$ is termed *aperiodic* (Bullo et al., 2009).

**Hypergraphs**    A hypergraph is a generalization of a graph in which an edge can connect any number of vertices. A hypergraph can be denoted by a triple $\mathcal{H}[\boldsymbol{W}^h] = \{\mathcal{V}, \mathcal{E}, \boldsymbol{W}^h\}$. To avoid

notation confusion, we still use $\mathcal{V}$ for the set of nodes and $\mathcal{E}$ for the set of hyperedges. We set $|\mathcal{V}| = N$ and $|\mathcal{E}| = M$, and $\mathcal{E}(i)$ denotes a set containing all the nodes sharing at least one hyperedge with node $i$. Usually, $\boldsymbol{W}^h$ is a diagonal matrix for hyperedges, where $W_{ee}^h$ represents the weight of the hyperedge $e$. In this paper, we extend the weight representation to a triple tensor $\boldsymbol{W}^h \in \mathbb{R}^{N \times N \times M}$, where $w_{i,j}^e$ designates an element in $\boldsymbol{W}^h$. The incidence matrix $\boldsymbol{H} \in \mathbb{R}^{N \times M}$ defines $H_{i,e} = 1$ if the node $i$ belongs to the hyperedge $e$, otherwise $H_{i,e} = 0$. We generalize the indicator within $\boldsymbol{W}^h$ by setting $w_{i,j}^e$ as nonzero if node $i, j$ are connected by a hyperedge $e$, otherwise 0.

**Neural Message Passing**  Neural Message Passing (MP; Gilmer et al. (2017)) stands as the prevailing propagator for updating node representations in GNNs. We denote $\boldsymbol{x}_i^{(k-1)}$ as the features of node $i$ in layer $(k-1)$ and $a_{j,i} \in \mathbb{R}^d$ as the edge features from node $j$ to node $i$. An MP layer reads

$$\mathbf{x}_i^{(k)} = \gamma^{(k)} \left( \mathbf{x}_i^{(k-1)}, \square_{j \in \mathcal{N}_i} \phi^{(k)} \left( \mathbf{x}_i^{(k-1)}, \mathbf{x}_j^{(k-1)}, a_{j,i} \right) \right), \tag{1}$$

where $\square$ denotes a differentiable, (node) permutation invariant function, such as summation, mean, or maximization. The $\gamma$ and $\phi$ denote differentiable functions such as MLPs (Multi-Layer Perceptrons), and $\mathcal{N}_i$ represents the set of one-hop neighbors of node $i$. The MP mechanism updates the feature of each node by aggregating their self-features with neighbors' features.

The classic MPs can be extended to hypergraphs that to consider interactions among multiple nodes reflected in a hyperedge. At the $k$th layer:

$$\mathbf{x}_i^{(k+1)} = \Psi^{(k)} \left( \mathbf{x}_i^{(k)}, \Phi_{1, e \in \mathcal{E}(i)} \left( e, \Phi_{2, j \in e}^{(k)} (\{\mathbf{x}_j^{(k)}\}, \{a_{j,i}^e\}) \right) \right), \tag{2}$$

where $\Phi_1^{(k)}$ denotes a differentiable, (hyperedge) permutation-invariant function, and $\Phi_2^{(k)}$ is a differentiable, (node) permutation invariant function. $\Psi^{(k)}$ denotes another differentiable function of propagation, and $j \in e$ implies $H_{j,e} = 1$ or $a_{j,i}^e \neq 0$.

## 3 CONNECTING OPINION DYNAMICS WITH MESSAGE PASSING

### 3.1 FRENCH-DEGROOT MODEL

The French-DeGroot (FD), originally introduced by French Jr (1956) and later developed by Harary (1959), Norman et al. (1965) and DeGroot (1974), is a groundbreaking agent-based model that simulates the evolution of opinions. In a given population of $N$ individuals, each individual holds an opinion $\boldsymbol{x}_i(k)$ at discrete time instances $k = 0, 1, \cdots$. The evolution of an individual's opinion is

$$\boldsymbol{x}_i(k+1) = \sum_{j=1}^{N} w_{ij} \boldsymbol{x}_j(k), \tag{3}$$

where the non-negative *influence weight* $w_{ij}$ satisfying $\sum_{j=1}^{N} w_{ij} = 1$. If $w_{ij} > 0$, individuals $i$ and $j$ are neighbors. The influence weight signifies the relative impact that $j$ exerts on $i$ during each opinion update. Importantly, all individuals concurrently update their opinions at each time step. The FD model captures how individual opinions converge within a group, potentially leading to consensus, resembling an opinion pooling process. It can be interpreted as an MP, where the graph represents a community with each node representing an individual, emulating how information is exchanged within a specific type of neural network.

**Convergence Analysis**  A fundamental result regarding the convergence of the FD model is well-established, demonstrating that consensus is achieved exponentially fast for a strongly connected and aperiodic graph. This result can be found in references such as Ren & Beard (2008); Proskurnikov & Tempo (2017); Bullo et al. (2009); Ye (2019).

**Proposition 1.** *Consider the evolution of opinions $\boldsymbol{x}_i(k)$ for each individual $i$ within the network $\mathcal{G}[\mathbf{W}]$ according to (3). Assuming that $\mathcal{G}[\mathbf{W}]$ is strongly connected and aperiodic, and that $\mathbf{W}$ is row-stochastic. Define $\zeta$ as the dominant left eigenvector of $\mathbf{W}$, then $\lim_{k \to 0} \mathbf{x}(k) = (\zeta^\top \mathbf{x}(0)) \mathbf{1}_N$ exponentially fast.*

It's worth noting that any graph with a self-loop is considered aperiodic, implying that exponential decay is likely to occur in graphs with relatively strong connectivity. Coincidentally, a similar phenomenon, known as *oversmoothing* (Nt & Maehara, 2019; Oono & Suzuki, 2019), has been explored

in the context of GNNs, where it is associated with the exponential decay of the *Dirichlet energy*, a measurement of the convergence degree of all features (weighted by graph structure). Despite originating from different fields, these two phenomena appear to describe similar processes.

**Connection to Neural Message Passing**   It's intriguing to observe that the FD model, often regarded as a micro-level model based on individuals simulating the evolution of individual opinions, shares similarities with a GNN model known as GRAND (Chamberlain et al., 2021). It describes a diffusion process on graphs by connecting heat conduction with MP. This connection is established through the discretization of a partial differential equation on graphs:

$$\frac{\partial}{\partial t}\boldsymbol{x}(t) = (\boldsymbol{A}(\boldsymbol{x}(t)) - \boldsymbol{I}_N)\boldsymbol{x}(t),\tag{4}$$

where $\boldsymbol{A}(\boldsymbol{x}(t))$ denotes the $N \times N$ attention matrix on nodes and $\boldsymbol{I}_N$ is an identity matrix. GRAND coincides with the FD model when $(\boldsymbol{A}(\boldsymbol{x}(t)) - \boldsymbol{I}_N)$ satisfies the row-stochastic property and a simple forward-Euler method is applied with a time step of one. This intriguing parallel between the two models highlights the interconnectedness of ideas in different domains of research.

## 3.2 Hegselmann-Krause Model

In the FD model, each agent possesses the capability to interact with any other agent, regardless of their opinions. However, in real-life scenarios, individuals typically engage in conversations primarily with those who share similar viewpoints. This fundamental aspect of human communication is accurately characterized and referred to as *bounded confidence* within the context of sociodynamics. The Hegselmann-Krause (HK) model (Rainer & Krause, 2002) defines bounded confidence as

$$\boldsymbol{x}_i(k+1) = \frac{1}{|\mathbb{B}(i, \boldsymbol{x}_i))|} \sum_{j \in \mathbb{B}(i, \boldsymbol{x}_i)} \boldsymbol{x}_j(k),\tag{5}$$

where $\mathbb{B}(i, \boldsymbol{x}_i) = \{j : \|\boldsymbol{x}_j(k) - \boldsymbol{x}_i(k)\| < \epsilon\}$ encompasses all individual $i$'s associated peers $j$, whose opinions diverge from individual $i$ within a confined region of radius $\epsilon_i \in \mathbb{R}$. This parameter represents the degree of uncertainty or tolerance within the model.

**Clustering and Oversmoothing in Heterophilious Dynamics**   The HK model demonstrates a clustering phenomenon driven by the self-alignment of agents. The number of clusters has been shown to have a negative correlation with the heterophily dependence among agents in a system (Motsch & Tadmor, 2014). Specifically, when interactions exhibit significant heterophily–meaning that agents tend to form stronger bonds with counterparts rather than with similar individuals–the dynamics tend to foster consensus. This tendency of individuals converging toward an 'environmental averaging' aligns with the oversmoothing issue in GNNs. One solution is to require an MP to retain at least two clusters at the end, where the Dirichlet energy is proven to have a lower bound. This can be achieved through techniques such as bi-clustering with repulsion (Fang et al., 2019; Jin & Shu, 2021; Wang et al., 2023). In the context of the HK model, it is advisable to avoid steep increases over compact supports when aggregating neighboring information.

## 4 ODNet: Opinion Dynamics-Inspired Neural Message Passing

Inspired by the mechanism of opinion dynamics, we introduce ODNet, a novel MP framework, employing the *influence function* $\phi(s)$ with bounded confidences. We offer a comprehensive interpretation of each component within ODNet, beginning with a discrete formulation and subsequently extending it to continuous forms that are applicable to both graphs and hypergraphs.

**Discrete Formation**   In the discrete domain, we formulate the update rule as follows:

$$\boldsymbol{x}_i(t+1) = \sum_{i=1}^{N} \phi(s_{ij})(\boldsymbol{x}_j(t) - \boldsymbol{x}_i(t)) + \boldsymbol{x}_i(t) + u(\boldsymbol{x}_i(t)),\tag{6}$$

where $\phi$ is a non-decreasing function of the similarity measure $s_{i,j}$ to node $i$ and node $j$, and $u(\boldsymbol{x}_i)$ is a control term for stability. For instance, $s_{ij}$ could be defined as the normalized adjacency matrix

(Kipf & Welling, 2017) or attention coefficients (Veličković et al., 2018). The monotonicity of $\phi$ characterizes the *influence weight* concerning node-node similarity. Our model opts for a piecewise $\phi$ function to delineate influence regions akin to bounded confidence. In a special case, with

$$\phi(s) = \begin{cases} \mu s, & \text{if } \epsilon_2 < s \\ s, & \text{if } \epsilon_1 \leq s \leq \epsilon_2 \\ 0, & \text{otherwise,} \end{cases} \tag{7}$$

our model (6) can be written as

$$\boldsymbol{x}_i(t+1) = \mu \sum_{\epsilon_2 < s_{i,j}} s_{i,j}(\boldsymbol{x}_j(t) - \boldsymbol{x}_i(t)) + \sum_{\epsilon_1 \leq s_{i,j} \leq \epsilon_2} s_{i,j}(\boldsymbol{x}_j(t) - \boldsymbol{x}_i(t)) + \boldsymbol{x}_i(t). \tag{8}$$

This formulation amplifies the influence weight for highly similar node pairs with coefficient $\mu > 0$ while cutting connections for node pairs with low similarity, resembling how individuals tend to ignore opinions beyond their bounded confidence.

Additionally, in certain extreme scenarios, individuals with significantly divergent opinions may exhibit hostile attitudes toward each other. To model such instances, we consider:

$$\phi(s) = \begin{cases} \mu s, & \text{if } \epsilon_2 < s \\ s, & \text{if } \epsilon_1 \leq s \leq \epsilon_2 \\ \nu(1-s), & \text{otherwise,} \end{cases} \tag{9}$$

where $\mu > 0$ and $\nu < 0$. In the context of GNNs, it allows not only learning from *positive* neighbors with similarity but also extracting *negative* information from nodes with discrepancies. It's worth noting that the negative coefficient $\nu$ implies that some node pairs consistently repel each other, potentially causing undesirable system dilation. Therefore, a control term $u(x_i)$ is introduced for system stability. A simple approach is to design a potential function $P(x)$ where $\nabla P(x) \to \infty$ as $x \to \infty$, and set $u(x_i) = \nabla P(x)|_{x=x_i}$. Various choices for $P(x)$ can be explored, as discussed by Kolokolnikov et al. (2011). The function $P(x)$ can be viewed as a moral constraint preventing individuals from resorting to extreme violence in conflict situations.

The different definitions of bounded confidence provided by (7) and (9) reflect the various behaviors of opinion exchange in a system, and these behaviors are linked to the different positions of the system along the opinion spectrum. While this can be conceptually determined by the intrinsic characteristics of the graph or the system, we propose the use of the *homophily level* (Pei et al., 2020) as an alternative quantitative measure. When passing messages on a specific graph, we recommend employing the former formulation for homophilic graphs and the latter for heterophilic graphs. Further investigations and explanations will be provided in Section 5.

**Continuous Formation**  In the realm of opinion dynamics, an individual's viewpoint typically undergoes gradual shifts rather than abrupt reversals. For instance, a person's political orientation is seldom confined to the extremes of either far-right or far-left, and an ultra-leftist rarely makes an overnight transition to a far-right position. Therefore, a natural refinement of the discrete MP model presented above is to generalize it into a continuous version. In broader terms, one can view a conventional MP model as a numerical discretization of the following continuous model:

$$\frac{\partial \boldsymbol{x}_i(t)}{\partial t} = \sum_{i=1}^{N} \phi(s_{ij})(\boldsymbol{x}_j(t) - \boldsymbol{x}_i(t)) + u(\boldsymbol{x}_i) \tag{10}$$

The continuous formulation (10) outlined above is amenable to various numerical approximation techniques corresponding to a discrete model with a specific residual compensation scheme. Consequently, a range of Ordinary Differential Equation (ODE) solvers can be employed for ODNET, including Neural ODEs (Chen et al., 2018).

**Generalization on Hypergraphs**  Since ODNET constructs a general formulation for MP, it can be extended to hypergraphs as well. The primary distinction between graphs and hypergraphs lies in the fact that a hyperedge extends connectivity beyond the scope of traditional edges. This extension can be accommodated by generalizing the weight aggregation:

$$\frac{\partial \boldsymbol{x}_i}{\partial t} = \sum_{e:i \in e} \sum_{j \in e} \phi(s_{i,j}^e)(\boldsymbol{x}_j - \boldsymbol{x}_i) + u(\boldsymbol{x}_i). \tag{11}$$

Table 1: Average test accuracy on **homophilic** graphs over 10 random splits.

| Model | Cora | CiteSeer | PubMed | Coauthor CS | Computer | Photo |
|---|---|---|---|---|---|---|
| GCN (Kipf & Welling, 2017) | 81.5±1.3 | **71.9±1.9** | 77.8±2.9 | 91.1±0.5 | 82.6±2.4 | 91.2±1.2 |
| MoNet (Monti et al., 2017) | 81.3±1.3 | 71.2±2.0 | 78.6±2.3 | 90.8±0.6 | 83.5±2.2 | 91.2±2.3 |
| GraphSage-mean (Hamilton et al., 2017) | 79.2±7.7 | 71.6±1.9 | 77.4±2.2 | 91.3±2.8 | 82.4±1.8 | 91.4±1.3 |
| GraphSage-max (Hamilton et al., 2017) | 76.6±1.9 | 67.5±2.3 | 76.1±2.3 | 85.0±1.1 | N/A | 90.4±1.3 |
| GAT (Veličković et al., 2018) | **81.8±1.3** | 71.4±1.9 | **78.7±2.3** | 90.5±0.6 | 78.0±1.9 | 85.7±2.0 |
| GAT-PPR (Veličković et al., 2018) | 81.6±0.3 | 68.5±0.2 | 76.7±0.3 | 91.3±0.1 | **85.4±0.3** | 90.9±0.3 |
| CGNN (Xhonneux et al., 2020) | 81.4±1.6 | 66.9±1.8 | 66.6±4.4 | **92.3±0.2** | 80.3±2.0 | 91.4±1.5 |
| GDE (Poli et al., 2020) | 78.7±2.2 | 71.8±1.1 | 73.9±3.7 | 91.6±0.1 | 82.9±0.6 | **92.4±2.0** |
| GRAND-L (Chamberlain et al., 2021) | **83.6±1.0** | **73.4±0.5** | **78.8±1.7** | **92.9±0.4** | **83.7±1.2** | **92.3±0.9** |
| ODNet (ours) | **85.7±0.3** | **75.5±1.2** | **80.6±1.1** | **93.1±0.7** | **83.9±1.5** | **92.7±0.6** |

† The top three are highlighted by **First**, **Second**, **Third**.

| Model | Texas | Wisconsin | Cornell |
|---|---|---|---|
| MLP | 80.8±4.8 | 85.3±3.3 | 81.9±6.4 |
| GPRGNN (Chien et al., 2021) | 78.4±4.4 | 82.9±4.2 | 80.3±8.1 |
| H2GCN (Zhu et al., 2020) | **84.9±7.2** | **87.7± 5.0** | **82.7±5.3** |
| GCNII (Chen et al., 2020) | 77.6±3.8 | 80.4±3.4 | 77.9±3.8 |
| Geom-GCN (Pei et al., 2020) | 66.8±2.7 | 64.5±3.7 | 60.5±3.7 |
| PairNorm (Zhao & Akoglu, 2020) | 60.3±4.3 | 48.4±6.1 | 58.9±3.2 |
| GraphSAGE (Hamilton et al., 2017) | 82.4±6.1 | 81.2±5.6 | 76.0±5.0 |
| GAT (Veličković et al., 2018) | 52.2±6.6 | 49.4±4.1 | 61.9±5.1 |
| GCN (Kipf & Welling, 2017) | 55.1±5.2 | 51.8±3.1 | 60.5±5.3 |
| GraphCON (Rusch et al., 2022) | **85.4±4.2** | **87.8±3.3** | **84.3±4.8** |
| ODNet | **88.3±3.2** | **89.1± 2.9** | **86.5±5.5** |

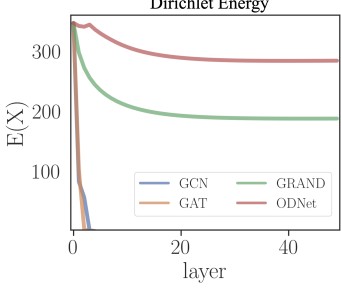

Figure 2: Decays of Dirichlet energy with layers on **Texas**.

Table 2: Average test accuracy on **heterophilic** graphs.

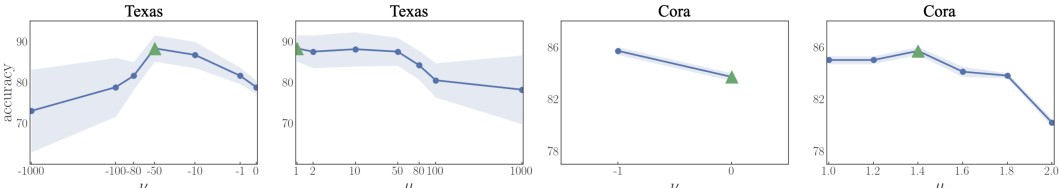

Figure 3: Impact of scaling factors $\nu$ and $\mu$ on **Texas** and **Cora**.

This formulation aligns with the notion that in large communities, information propagates through smaller sections (nodes that share a hyperedge) rather than through individual pairwise interactions. Similar to the graph case, the choice of $s_{i,j}^e$ may vary, such as attention coefficients (Bai et al., 2021).

**Remark 1.** *The collective behaviors in HK-driven modeling mitigate the oversmoothing issue for continuous MP schemes on hypergraphs. See Appendix A for further discussions.*

## 5 Graph Representation Learning

### 5.1 Experimental Protocol

**Benchmark Datasets** This section validates the efficacy of ODNet through classic node-level representation learning tasks on a variety of datasets spanning three types of graphs, including six homophilic graphs (**Cora** (McCallum et al., 2000), **Citeseer** (Sen et al., 2008), **Pubmed** (Namata et al., 2012), **Coauthor CS** (Shchur et al., 2018), **Computer** (Namata et al., 2012), and **Photo** (Namata et al., 2012)), three heterophilic graphs (**Texas**, **Wisconsin**, and **Cornell** from the WebKB dataset (García-Plaza et al., 2016)), and four hypergraphs based on the citation network (Yadati et al., 2019). For additional descriptions, please refer to Appendix B.1.

**Training Setup** We compare our model to a diverse set of top-performing baseline GNN models, including classic graph convolutions, MPs with continuous updating schemes, and the latest hypergraph models. For ODNet, we trained the model using a neural ODE solver with Dormand–Prince

Table 3: Average test accuracy on **hypergraphs** over 10 random splits.

| Model | Cora-coauthor | Cora-cocitation | CiteSeer-cocitation | PubMed-cocitation |
|---|---|---|---|---|
| HGNN (Feng et al., 2019) | 82.6±1.7 | **79.4±1.4** | 72.5±1.2 | 86.4±0.4 |
| HYPERGCN (Yadati et al., 2019) | 79.5±2.1 | 78.5±1.3 | 71.3±0.8 | 82.8±8.7 |
| HCHA (Bai et al., 2021) | 82.6±1.0 | 79.1±1.0 | 72.4±1.4 | 86.4±0.4 |
| HNHN (Dong et al., 2020) | 77.2±1.5 | 76.4±1.9 | 72.6±1.6 | 86.9±0.3 |
| UNIGCNII (Huang & Yang, 2021) | **83.6±1.1** | 78.8±1.1 | 73.0±2.2 | 88.3±0.4 |
| HYPERND (Tudisco et al. (2021) | 80.6±1.3 | 79.2±1.1 | 72.6±1.5 | 86.7±0.4 |
| ALLDEEPSETS (Chien et al., 2022) | 82.0±1.5 | 76.9±1.8 | 70.8±1.6 | **88.8±0.3** |
| ALLSETTRANSFORMER (Chien et al., 2022) | 83.6±1.5 | 78.6±1.5 | **73.1±1.2** | 88.7±0.4 |
| ED-HNN (Wang et al., 2022) | **84.0±1.6** | **80.3±1.4** | **73.7±1.4** | **89.0±0.5** |
| ODNET (Ours) | **84.5±1.6** | **80.7±0.9** | **74.0±0.9** | **89.0±0.4** |

adaptive step size scheme (DOPRI5). In homophilic datasets, we utilized 10 random weight initializations and random splits, with each combination randomly selecting 20 instances for each class. In heterophilic and hypergraph datasets, we used the fixed 10 training/validation splits by Pei et al. (2020) and Yadati et al. (2019), respectively. For further details, please refer to Appendix C.

## 5.2 NODE CLASSIFICATION

**Graphs** Tables 1-2 present the average accuracy for predicting node labels in both homophilic and heterophilic graphs. ODNET consistently ranks among the top-performing methods with minimal variance. The performance results for baseline methods are sourced from prior studies (Chamberlain et al., 2021; Chien et al., 2021; Wang et al., 2022). Notably, our model outperforms other continuous MP techniques, such as GRAND, by introducing the bounded confidence mechanism and the respective influence weights. This superiority is particularly evident on heterophilic graphs, where the repulsive force among dissimilar node pairs significantly enhances prediction accuracy. Furthermore, Figure 3 illustrates the distinct preferences of the influence function for homophilic and heterophilic graphs. We recommend following (7) for the former and (9) for the latter in general. The similarity cutoff also exhibits differing preferences. In homophilic graphs, nodes tend to amplify attraction among similar entities, while in heterophilic graphs, dissimilar nodes benefit more from emphasizing repulsion. Additional evidence is provided in Appendix D.1.

**Hypergraphs** In contrast to graph data with relatively sparse connections, hypergraphs utilize a few hyperedges and establish densely connected local communities. As reported in Table 3, ODNET consistently outperforms most baseline methods with a significant improvement. The only exception is ED-HNN, where our method achieves a slightly less pronounced advantage. It is worth noting that our ODNET adopts the hypergraph weights $a_{ij}^e$ from HGNN (Feng et al., 2019) with a simple Euler scheme of first-order forward differences. In contrast, ED-HNN employs the second-order difference, which intrinsically contains more comprehensive and expressive information. In this case, our method demonstrates great potential for significantly enhancing the performance of a basic method with minimal additional complexity, surpassing even the most advanced methods.

## 5.3 DIRICHLET ENERGY, OVERSMOOTHING, AND COMMUNITY CONSENSUS

Many MP methods encounter the issue of oversmoothing, limiting their ability to enable deep networks to achieve expressive propagation. As a common metric, a GNN model is considered to alleviate the oversmoothing problem if its Dirichlet energy rapidly approaches a lower bound as the number of network layers increases (Cai & Wang, 2020). Figure 2 illustrates the decay of Dirichlet energy on **Texas** with all network parameters randomly initialized. The two conventional MPs, GCN and GAT, exhibit a sudden progression of Dirichlet energy with exponential decay. In contrast, GRAND employs a small multiplier to delay all nodes' features to collapse to the same value. ODNET stabilizes the energy decay with bounded confidence and the influence weights, offering a simple and efficient solution to alleviate the oversmoothing issue. Since stable Dirichlet energy reflects the disparity of feature clusters, the observation that a decreasing profile of $\phi$ reduces Dirichlet energy under stable conditions is consistent with simulation results in opinion dynamics that heterophily dynamics enhances consensus (Motsch & Tadmor, 2014). Here 'heterophily' signifies the tendency of a graph to form stronger connections with those who are different rather than those who are similar, which is a different concept from the 'heterophilic graph' in GNNs.

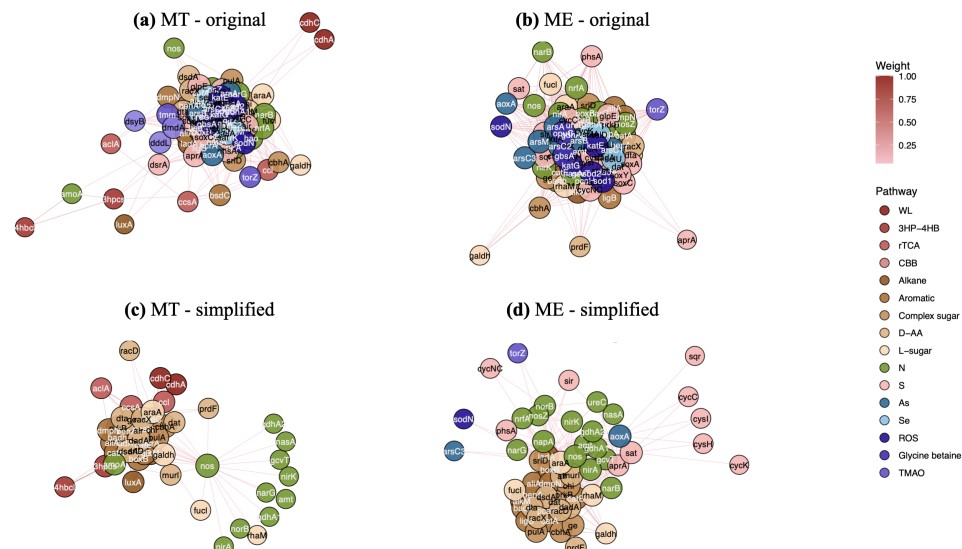

Figure 4: Co-occurrence network of selected metabolic genes in ME or MT before (a-b) and after (c-d) being simplified by ODNET. Connections are considered strong with weights $> 0.05$, and genes without a strong connection with any other peers are removed.

## 6   SOCIAL NETWORK ARCHITECTURE SIMPLIFICATION

Microorganisms are the most extensively distributed and numerous group on Earth, which thrive in a wide array of moderate and extreme environments, such as deep-sea hydrothermal vents, ocean trenches, and plateaus (Shu & Huang, 2022). The remarkable diversity among microorganisms finds its primary expression through their intricate metabolic pathways (Louca et al., 2018; Coelho et al., 2022). Consequently, investigating the connections between microbial metabolism in distinct environments carries profound significance in unraveling the intricate interplay between Earth's diverse ecosystems and the lives inhabiting them. Metagenomic analyses have revealed the remarkable complexity inherent to metabolic gene networks, due to the diversity and richness of functional genes and their interconnections. It thus becomes a necessity to simplify metabolic gene networks for investigating relationships among functional genes and key genes. Presently, the prevailing approach involves adjusting connection weights to streamline the network, often relying on the biological expertise (Liu et al., 2022). However, the absence of a standardized simplification criterion results in a heavy bias in network structures influenced by the subjective opinions of biologists.

**Problem Formulation and Training Setup**   As an example of the environmental microbiome analysis, the co-occurrence network is challenging to interpret due to the massive and complicated characteristics of genes and the unclear standard for assessment. The target here is to learn meaningful influence weights between gene pairs that simplify the co-occurrence network with effective biological justification. To this end, two networks originated from the microbial comparison between the *Mariana Trench* (MT) and *Mount Everest* (ME) networks (Liu et al., 2022) are utilized, where nodes are functional genes and edges are weighted by the probability of two key functional gene sets simultaneously occurring in the same species. Edges with exceptionally small weights will be discarded as noisy observations. As we are eager to identify the key genes and gene clusters from gene interactions, we construct the graph with initial connectivity (edges and edge weights), leaving any node attributes (*e.g.*, function, pathway) unobserved. We define a node-level classification task for predicting whether a node is a 'strong', 'medium', or 'weak' influencer to its community, where the three levels are cut by their degree. Further details are provided in Appendix E.

**Result Analysis**   We trained two independent ODNETs on ME and MT networks, which achieved prediction accuracy as high as 96.9% and 75.0%, respectively. For both networks, metabolic genes were classified based solely on topological information, without the introduction of any a priori node features. Figure 4 visualizes the two networks in their original and the simplified appearance, respectively. For all the networks, an edge weight cutoff of 0.05 was applied to eliminate weak

connections that could not be distinguishable from background noise. The original network without any simplification appeared cluttered and difficult to interpret (Figure 4a-b). In contrast, the simplified networks greatly enhanced the readability of the co-occurrence network while retaining reasonable biological significance (Figure 4c-d). Furthermore, the simplified network was able to identify the biologically key genes that acted as "opinion leaders", serving as bridges connecting different metabolic pathways. For example, in the MT network, the key gene nitrous oxide reductase (nos) bridged the carbon (Alkane, Aromatic, Complex sugar, D-AA and L-sugar) and nitrogen metabolism (N) in Figure 4c, whereas in the ME network, the key genes of sulfate reduction (sat and aprA) coupled the carbon and sulfur metabolism (S) (Figure 4d). Thus, ODNET could be employed to present more discernible networks in environmental microbiome studies, and aid in comprehending key metabolic functions within microbiomes from diverse environments.

## 7 RELATED WORK

**Neural Message Passing on Graphs and Hypergraphs**   Neural message passing establishes a general computational rule for updating node representations in attributed graphs (Gilmer et al., 2017; Battaglia et al., 2018; Hamilton, 2020). This framework has seen active extensions into continuous graph convolutions (Poli et al., 2020; Brandstetter et al., 2021; Chamberlain et al., 2021; Liu et al., 2023; Wang et al., 2023). Notably, SINN (Okawa & Iwata, 2022) explored the possible combination of opinion dynamics and MP with a different implementation from ours. We employed a potential term to avoid excessive attraction or repulsion from neighbors and circumvent the oversmoothing issue. Also, we established bounded confidence with adjustable scaling factors to balance the attractive or repulsive forces when aggregating the neighborhood information. In parallel, Feng et al. (2019); Gao et al. (2022) extended GCN and established a general convolution framework employing the incidence matrix for hypergraph learning. Various techniques have also undergone expansion, such as the attention mechanism (Bai et al., 2021), spectral theory (Yadati et al., 2019), and node potential (Wang et al., 2022).

**Collective Dynamics**   In classical opinion dynamics systems, a first-order formulation of information exchange is typically employed, relying on the positions of individuals. This formulation naturally connects with a second-order formulation consistent with Newtonian dynamics, which finds applications in phenomena like animal flocking, cell clusters, and self-organizing particles (Holm & Putkaradze, 2006; Carrillo et al., 2010a; Kolokolnikov et al., 2013). These scenarios fall under the purview of *collective dynamics*, wherein agents move together based on attraction and repulsion forces (D'Orsogna et al., 2006; Carrillo et al., 2010b; Motsch & Tadmor, 2014; Carrillo & Shu, 2023). For example, the Cucker-Smale model (Cucker & Smale, 2007) extends the HK model to a second-order formulation involving both position and velocity; Fang et al. (2019) investigated bi-cluster flocking with Rayleigh friction and attractive-repulsive coupling; Jin & Shu (2021) demonstrated a similar collective phenomenon with stochastic dynamics.

## 8 CONCLUSION

This study establishes intriguing connections between sociodynamics and graph neural networks, two distinct fields that both actively investigate social networks from different perspectives. By bridging concepts from these two fields, we introduce bounded confidence for neural message passing, a novel mechanism inspired by opinion dynamics. The proposed ODNET effectively addresses oversmoothing issues and consistently achieves top-notch performance in node prediction tasks across graphs with diverse local connectivity patterns. This success is attributed to the simplicity and efficacy of our piecewise message propagation rule. Moreover, our method showcases significant potential in simplifying complex real-world social networks, offering a fresh analytical approach that does not rely on existing attributive classification conventions.

The robust performance of ODNET extends its applicability to simplifying intricate networks containing a wealth of biological information, such as genes, gene-gene interactions, and metabolic pathways. This method's exceptional capacity to extract accurate insights and unveil the intrinsic mechanisms of cellular physiology provides invaluable support to biologists in deciphering the mechanisms of adaptation and pathway functions in the microbial realm. These findings are significant in understanding the interactions between Earth's environments and the metabolism of life.

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
