The Appendix is structured as follows:

- Appendix A extends the diffusion process on hypergraphs and justifies the collective behaviors in ODNET in effectively alleviating the oversmoothing issue encountered in hypergraph learning.

- Appendix B introduces benchmark datasets for node classification tasks.

- Appendix C specifies training setups for ODNET.

- Appendix D reports ablation study and additional experimental results of ODNET.

- Appendix E introduces more problem setup for the learning tasks on the co-occurrence gene network.

- Appendix F supplements additional biological backgrounds for understanding the co-occurrence network we assessed in Section 6.

## A  SCALE TRANSLATION OF HYPERGRAPH: DIFFUSION AND PARTICLE DYNAMICS

In Section 4, we derived diffusion-type dynamics based on collective behaviors for hypergraphs. It is also possible to incorporate diffusion-based models with macroscopic interpretations, which only consider attractions between individuals or agents.

### A.1  HYPERGRAPH DIFFUSION

Consider the node feature space $\Omega = \mathbb{R}^d$ and the tangent vector field space $T\Omega = \mathbb{R}^d$. For $\mathbf{x}, \mathbf{y} \in \Omega$ and $\mathfrak{x}, \mathfrak{y} \in T\Omega$, where $\mathfrak{x}_{i,j} = -\mathfrak{x}_{j,i}$, we adopt the following inner products:

$$\langle \mathbf{x}, \mathbf{y} \rangle = \sum_{i,j} \mathbf{x}_i \mathbf{y}_j, \quad [\mathfrak{x}, \mathfrak{y}] = \sum_{i>j} \sum_{e \in \mathcal{E}} h_{i,j}^e \, \mathfrak{x}_{i,j} \mathfrak{y}_{i,j}. \tag{12}$$

Here $h_{i,j}^e$ represents a tuple related to node $i, j$ and hyperedge $e$, and $h_{i,j}^e = 0$ if $H_{i,e} H_{j,e} = 0$. We set $h_{i,j}^e$ to satisfy $\sum\limits_{j} \sum\limits_{e \in \mathcal{E}} h_{i,j}^e = 1$. For any $\mathfrak{u} \in T\Omega$,, by the adjoint relation:

$$[\mathfrak{u}, \nabla \mathbf{x}] = \langle \mathbf{x}, \mathrm{div}\mathfrak{u} \rangle,$$

where $\nabla \mathbf{x} = \mathbf{x}_j - \mathbf{x}_i$, we derive:

$$(\mathrm{div}\mathfrak{u})_j = \sum_{i} \sum_{e \in \mathcal{E}} h_{i,j}^e u_{i,j}. \tag{13}$$

This leads to a formal diffusion process of a hypergraph:

$$\frac{d\mathbf{x}_i}{dt} = \mathrm{div}\nabla \mathbf{x}_i = \sum_{j} \sum_{e \in \mathcal{E}} h_{i,j}^e (\mathbf{x}_j - \mathbf{x}_i). \tag{14}$$

For simplicity, we rewrite (14) as

$$\frac{d\mathbf{x}}{dt} = -\mathcal{L}\mathbf{x}, \tag{15}$$

where $\mathcal{L} = I - (\sum\limits_{e \in \mathcal{E}} h_{i,j}^e)$ is a hypergraph operator. When $\mathcal{L}$ is semi-positive definite (s.p.d.), we define (14) as a diffusion-type process of a hypergraph. The different choices of $h_{i,j}^e$ lead to diverse diffusion-type equations. For example, when we take forward Euler discretization on (14) and use the matrix

$$(\sum_{e \in \mathcal{E}} h_{i,j}^e) = D_v^{-\frac{1}{2}} H W D_e^{-1} H^T D_v^{-\frac{1}{2}},$$

we obtain a simplified HGNN without channel mixing.

## A.2 Oversmoothing Analysis on Hypergraph Diffusion

In the context of diffusion-type hypergraph networks, we define the Dirichlet energy of a hypergraph $\mathcal{H}$ of vector field $\mathbf{x} \in \mathbb{R}^{N \times d}$ as

$$\mathbf{E}(\mathbf{x}) := \sum_{i,j=1}^{N} \sum_{e \in \mathcal{E}} H_{i,e} H_{j,e} \|\mathbf{x}_i - \mathbf{x}_j\|^2. \tag{16}$$

**Remark 1.** *It's worth noting that, on hypergraphs, the Dirichlet Energy can also be defined as* $\mathbf{E}(\mathbf{x}) := \mathrm{tr}(\mathbf{x}^\top \mathcal{L} \mathbf{x})$ *associated with the graph Laplacian* $\mathcal{L}$*. However, for simplicity and because* $\mathcal{L}$ *is not a deterministic matrix, we adopt a more straightforward definition as used in previous work (Rusch et al., 2022). This simplification allows us to effectively capture the differences among node features, making it an acceptable choice.*

Furthermore, we define oversmoothing as follows:

**Definition 1.** *Let* $\mathbf{x}^l$ *denote the hidden features of the* $l^{th}$ *layer. We define oversmoothing in a hypergraph neural network as the exponential convergence to zero of the layer-wise Dirichlet energy as a function of* $l$*, i.e.,*

$$\mathbf{E}(\mathbf{x}^l) \leq C_1 e^{-C_2 l}, \tag{17}$$

*where* $C_1$ *and* $C_2$ *are positive constants.*

Our analysis reveals that oversmoothing is a common issue in hypergraph diffusion networks, as $|\mathbf{x}| \leq C e^{-\gamma t}$, where $\gamma$ is the smallest positive eigenvalue of $\mathcal{L}$. This oversmoothing arises due to the diffusion structure, as node features $\mathbf{x}$ decay exponentially to zero under an s.p.d kernel $\mathcal{L}$.

**Connection to Particle Dynamics**  It is noteworthy that there is a striking similarity between (14) and self-organized dynamics in particle systems (Motsch & Tadmor, 2014). In this context, rather than a mere discretization of diffusion on hypergraphs, (14) represents a particle dynamics scenario where $h_{i,j}^e$ signifies the interactive force between nodes $i, j$ under a specific field $e$. Equation (14) corresponds to a particular case of (11) where only attractive forces influence the message evolution. However, this assumption is not universal for particle systems, and as demonstrated in Section A.2, it can lead to oversmoothing issues.

## A.3 Final Justification

Oversmoothing has emerged as a well-recognized concern in many MP schemes, particularly when applied to hypergraphs characterized by denser local connections than traditional graphs. Therefore, addressing the oversmoothing issue becomes of great importance in the design of propagation rules for hypergraph networks.

While it is conceivable to extend a GRAND-like framework (Chamberlain et al., 2021) to hypergraphs with macroscopic interpretations, these diffusion-type dynamics at the macro level are susceptible to oversmoothing of feature evolution, similar to traditional GNNs. Alternatively, as we discussed in Section 4, the HK model can be interpreted as a diffusion process on graphs featuring piecewise attraction and repulsion behaviors, as outlined in (10) and (11). When devising MP aggregation rules, employing a comprehensive framework for microscopic models based on collective behaviors within a complete interaction system proves effective in mitigating the oversmoothing issue.

## B  Benchmark Datasets for Node Classification

### B.1  Benchmark Datasets

**Graphs**  We consider two types of homophilic and heterophilic graphs. These categorizations are based on the concept of *homophily level* introduced by Pei et al. (2020):

$$\mathcal{H} = \frac{1}{|V|} \sum_{v \in V} \frac{\text{Number of } v\text{'s neighbors who have the same label as } v}{\text{Number of } v\text{'s neighbors}}.$$

Table 4 provides an overview of the statistical information for the six homophilic graphs and three heterophilic graphs, along with their respective homophily levels. A low homophily level indicates that the dataset leans more towards being heterophilic, where most neighbors do not share the same class as the central node. Conversely, a high homophily level signifies that the dataset tends towards homophily, with similar nodes more likely to be interconnected. The datasets considered in Section 5 encompass a wide range of homophily levels to guarantee a diverse set of scenarios for evaluation.

Table 4: Summary of **graph** datasets used in experiments.

| Dataset | # classes | # features | # nodes | # edges | homophily level |
|---|---|---|---|---|---|
| **Cora** | 7 | 1,433 | 2,708 | 5,429 | 0.83 |
| **CiteSeer** | 6 | 3,703 | 3,327 | 4,732 | 0.71 |
| **PubMed** | 3 | 500 | 19,717 | 44,338 | 0.79 |
| **CoauthorCS** | 15 | 6,805 | 18,333 | 100,227 | 0.80 |
| **Computer** | 10 | 767 | 13,381 | 245,778 | 0.77 |
| **Photo** | 8 | 745 | 7,487 | 119,043 | 0.83 |
| **Texas** | 5 | 1,703 | 183 | 309 | 0.11 |
| **Wisconsin** | 5 | 1,703 | 251 | 499 | 0.21 |
| **Cornell** | 5 | 1,703 | 183 | 295 | 0.30 |

**Hypergraphs** The hypergraph variant of ODNET undergoes an evaluation through semi-supervised node classification tasks conducted on four benchmark hypergraphs extracted from citation networks. For co-citation networks (**Cora-cocitation**, **CiteSeer-cocitation**, and **PubMed-cocitation**), documents cited by a given document are interconnected by a hyperedge. Similarly, the co-authorship networks (**Cora-coauthor**) aggregates all documents co-authored by an individual into a single hyperedge. For further elaboration and in-depth details, we encourage interested readers to refer to the work by Yadati et al. (2019).

Table 5: Summary of **hypergraph** datasets used in experiments.

| Dataset | # classes | # features | # hypernodes | # hyperedges | avg. hyperedge size |
|---|---|---|---|---|---|
| **Cora-coauthor** | 7 | 1,433 | 2,708 | 1,072 | 4.2±4.1 |
| **Cora-cocitation** | 7 | 1,433 | 2,708 | 1,579 | 3.0±1.1 |
| **CiteSeer-cocitation** | 6 | 3,703 | 3,312 | 1,079 | 3.2±2.0 |
| **PubMed-cocitation** | 3 | 500 | 19,717 | 7,963 | 4.3±5.7 |

## C  TRAINING SETUP FOR ODNET

All implementations are programmed with `PyTorch-Geometric` (version 2.0.1) (Fey & Lenssen, 2019) and `PyTorch` (version 1.7.0) and run on NVIDIA® Tesla A100 GPU with $6,912$ CUDA cores and 80GB HBM2 mounted on an HPC cluster. All the details to reproduce our results have been included in the submission. The program will be publicly available upon acceptance.

For common hyperparameters, such as learning rate and weight decay, we used Ray Tune with a hundred trials using an asynchronous hyperband scheduler with a grace period of 50 epochs. The tuning space is reported in Table 6 For homophilic datasets, we use 10 random splits, with each combination randomly selecting 20 numbers for each class. For heterophilic data, we use the original fixed 10 split datasets. The optimal combination of hyper-parameters are reported in Table 7.

## D  ADDITIONAL INVESTIGATION

### D.1  INFLUENCE FUNCTION

In this section, we delve into the impact of different selections of the influence function $\phi$ on the performance of ODNET. This investigation encompasses various aspects, including the choice of

Table 6: Hyperparameter Search Space

| Hyperparameters | Search Space | Distribution |
|---|---|---|
| learning rate | $[10^{-6}, 10^{-1}]$ | log-uniform |
| weight decay | $[10^{-3}, 10^{-1}]$ | log-uniform |
| dropout rate | $[0.1, 0.8]$ | uniform |
| hidden dim | $\{64, 128, 256\}$ | categorical |
| $\beta$ | $[0, 1]$ | uniform |

Table 7: Optimal setting of hyperparameters in reproducing the results in Section 5.

| Dataset | $\epsilon_1$ | $\epsilon_2$ | time (T) | $\nu$ | $\mu$ |
|---|---|---|---|---|---|
| **Cora** | 0.012 | 0.40 | 12 | 0 | 1.4 |
| **CiteSeer** | 0.01 | 0.90 | 10 | 0 | 3.0 |
| **PubMed** | 0.01 | 0.40 | 20 | 0 | 2.2 |
| **CoauthorCS** | 0.01 | 0.40 | 15 | 0 | 1.7 |
| **Computer** | 0.01 | 0.50 | 15 | 0 | 5.0 |
| **Photo** | 0.01 | 0.40 | 12 | 0 | 10.0 |
| **Texas** | 0.50 | 0.80 | 12 | -50 | 1.0 |
| **Wisconsin** | 0.60 | 0.80 | 12 | -10 | 2.0 |
| **Cornell** | 0.12 | 0.40 | 12 | 0 | 2.0 |
| **Cora-coauthor** | 0 | 1 | 0.1 | 1.0 | 1.0 |
| **Cora-cocitation** | 0 | 1 | 0.1 | 1.0 | 1.0 |
| **PubMed-cocitation** | 0 | 1 | 0.1 | 1.0 | 1.0 |
| **CiteSeer-cocitation** | 0 | 1 | 0.1 | 1.5 | 1.0 |

Table 8: Choices of influence function and scaling factors for **Cora**.

| $\epsilon_1$ | $\epsilon_2$ | $\mu$ | $\nu$ | **Accuracy** |
|---|---|---|---|---|
| 0.04 | 0.45 | 1.4 | 0 | 79.6±0.3 |
| 0.012 | 0.40 | 2.0 | 0 | 80.2±0.3 |
| 0.03 | 0.45 | 1.4 | 0 | 81.0±0.2 |
| 0.02 | 0.45 | 1.4 | 0 | 81.8±0.2 |
| 0.012 | 0.20 | 1.4 | 0 | 82.3±0.2 |
| 0.012 | 0.40 | 1.4 | -1 | 83.7±0.4 |
| 0.012 | 0.40 | 1.8 | 0 | 83.9±0.2 |
| 0.012 | 0.40 | 1.6 | 0 | 84.1±0.3 |
| 0.012 | 0.30 | 1.4 | 0 | 84.2±0.3 |
| 0.012 | 0.40 | 1.0 | 0 | 85.0±0.3 |
| 0.012 | 0.40 | 1.2 | 0 | 85.0±0.3 |
| 0.012 | 0.40 | 1.4 | 0 | 85.7±0.3 |
| 0.012 | 0.40 | 1.4 | 0 | 85.7±0.3 |
| 0.012 | 0.40 | 1.4 | 0 | 85.7±0.3 |
| 0.012 | 0.45 | 1.4 | 0 | 85.7±0.3 |

scaling factors ($\mu$ and $\nu$) and the definition of the similarity cutoffs ($\epsilon_1$ and $\epsilon_2$). Our findings are meticulously detailed in Table 8 and Table 9, with a particular focus on the homophilic graph (**Cora**) and the heterophilic graph (**Texas**), respectively. An interesting trend emerges concerning the parameter $\nu$, indicating a clear preference. Specifically, it is advisable to incorporate a repulsive effect on heterophilic graphs by assigning a negative value to $\nu$. Conversely, for homophilic graphs, where similarity plays a pivotal role, setting $\nu = 0$ is more appropriate.

To provide a direct comparison, Figure 5 showcases ODNET's performance under different similarity cutoffs, *i.e.*, $\epsilon_1$ and $\epsilon_2$. For **Cora**, we maintain $\mu = 1.4$ and $\nu = 0$, while for **Texas**, we set

Table 9: Choices of influence function and scaling factors for **Texas**.

| $\epsilon_1$ | $\epsilon_2$ | $\mu$ | $\nu$ | **Accuracy** |
|---|---|---|---|---|
| 0.50 | 0.80 | 2.0 | -1000 | 73.0±10.1 |
| 0.50 | 0.80 | 1000.0 | -50 | 78.2±8.4 |
| 0.50 | 0.80 | 2.0 | -100 | 78.8±7.2 |
| 0.50 | 0.80 | 2.0 | 0 | 78.8±1.6 |
| 0.50 | 0.80 | 100.0 | -50 | 80.5±4.2 |
| 0.50 | 0.60 | 1.0 | -50 | 81.0±3.0 |
| 0.70 | 0.80 | 1.0 | -50 | 81.1±4.2 |
| 0.50 | 0.80 | 2.0 | -80 | 81.6±3.4 |
| 0.50 | 0.80 | 2.0 | -1 | 81.6±2.0 |
| 0.60 | 0.80 | 1.0 | -50 | 86.5±3.5 |
| 0.50 | 0.90 | 1.0 | -50 | 86.5±3.0 |
| 0.50 | 0.70 | 1.0 | -50 | 87.0±3.4 |
| 0.50 | 0.80 | 2.0 | -10 | 86.7±3.2 |
| 0.40 | 0.80 | 1.0 | -50 | 87.0±3.0 |
| 0.50 | 0.80 | 2.0 | -50 | 87.6±4.0 |
| 0.50 | 0.80 | 2.0 | -50 | 87.6±4.0 |
| 0.50 | 0.80 | 10.0 | -50 | 88.1±4.2 |
| 0.50 | 0.80 | 1.0 | -50 | 88.3±3.2 |
| 0.50 | 0.80 | 1.0 | -50 | 88.3±3.2 |
| 0.50 | 0.80 | 1.0 | -50 | 88.3±3.2 |

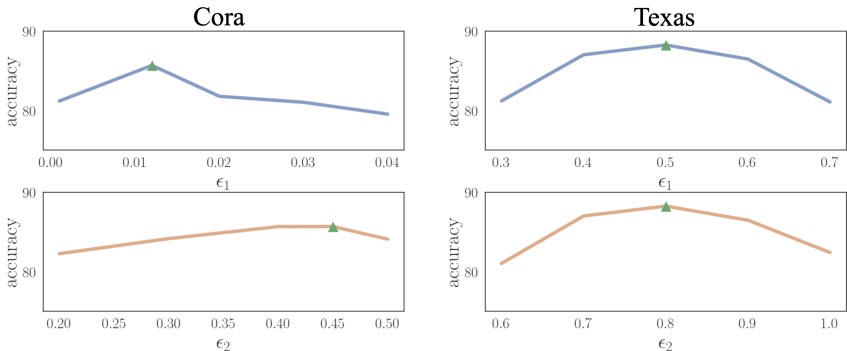

Figure 5: The impact of different $\epsilon_1$ and $\epsilon_2$ on ODNet.

$\mu = 1.0$ and $\nu = -50$. Generally, in the context of homophilic graphs, setting a relatively small value for $\epsilon_2$ tends to expand the region of nodes considered similar. This approach may be beneficial in mitigating the oversmoothing issue. Conversely, for heterophilic graphs, it is advisable to reverse information from only highly similar nodes, reflected in the choice of a larger $\epsilon_2$ value (up to $0.8$). However, it is crucial to exercise caution when pushing $\epsilon_2$ towards $1.0$, as a discernible reduction in performance becomes evident.

## D.2 NEURAL ODE SOLVERS

The Dormand–Prince adaptive step size scheme (DOPRI5) served as the neural ODE solver for ODNet. Additionally, we evaluated the performance of two other solvers across various datasets: the Runge-Kutta method (rk4) and the first-order Euler scheme (Euler). The outcomes are presented in Figure 6. Although different solvers did not consistently demonstrate a significant and sustained advantage of one over the others, our selection of DOPRI5 yielded the overall best results.

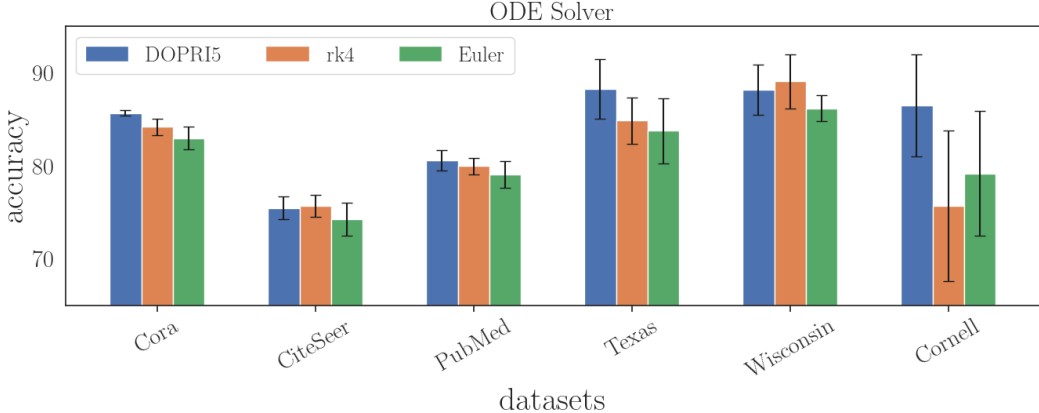

Figure 6: Prediction performance with different neural ODE solvers.

### D.3 EMBEDDING DYNAMICS

We employ the t-SNE algorithm to visualize the embedded features in a two-dimensional plane for the **Texas** dataset. We choose the output embeddings from the last layer at epochs $1, 10$, and $50$. As shown in Figure 7, an evident clustering trend becomes apparent as the number of training epochs increases. By the $50$th epoch, nodes with different labels are distinctly separable even in the reduced two-dimensional space.

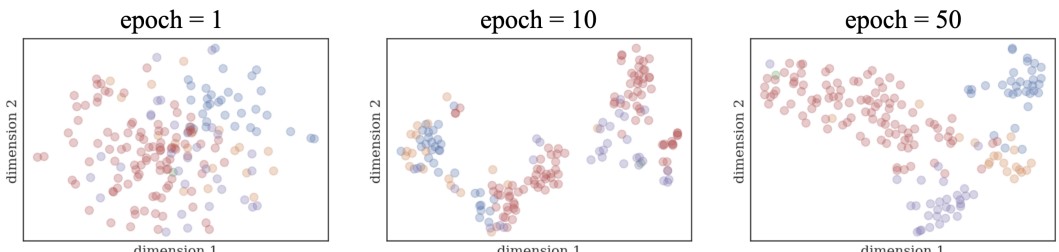

Figure 7: t-SNE visualization on node embeddings for **Texas** at epoch=$1, 10, 50$.

## E EXPERIMENTAL DETAILS FOR THE CO-OCCURRENCE NETWORK SIMPLIFICATION TASK

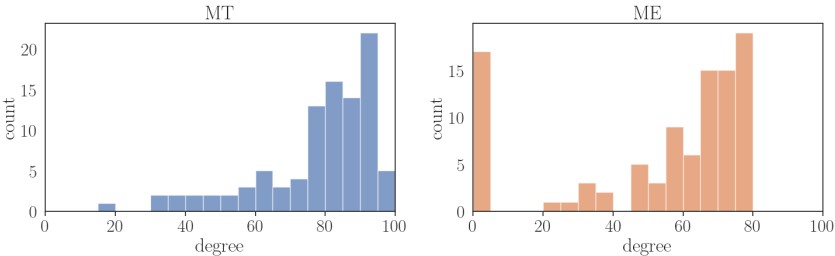

Figure 8: Distribution of node degree on **ME** and **MT** networks.

We established two distinct graphs for the Mariana Trench (MT) and Mount Everest (ME) gene co-occurrence networks, utilizing source data from Liu et al. (2022). In both networks, the nodes represent the same set of functional genes. The primary difference between them lies in the edge weights, which reflect unique co-occurrence patterns of gene pairs in MT and ME. This distinction

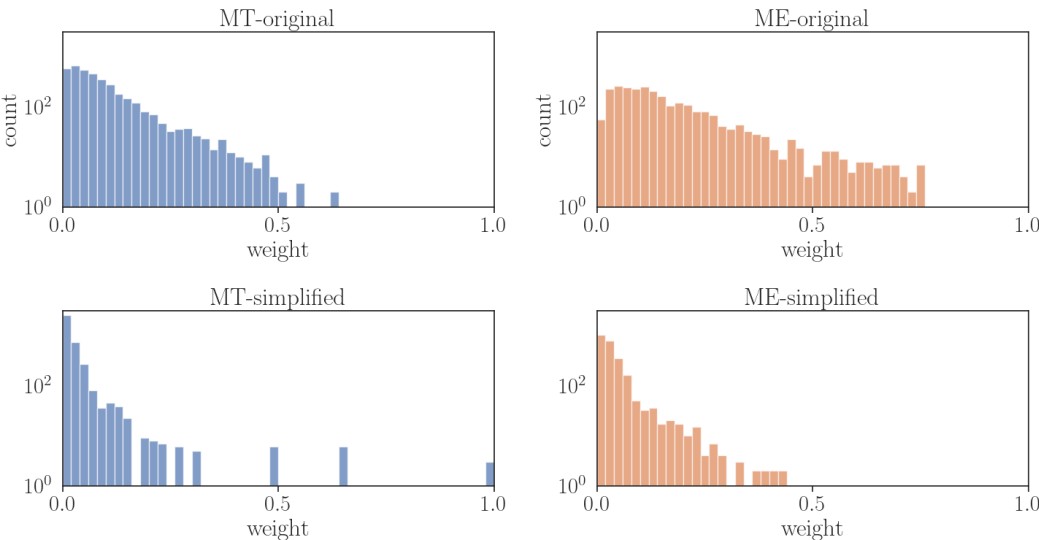

Figure 9: Distribution of edge weights on **ME** and **MT** networks.

is visually evident when comparing the top two histograms in Figure 9, illustrating the varying distributions of gene influence weights in MT and ME graphs.

To construct these graphs, we connected all node pairs with non-zero edge weights, resulting in a total of $2,517$ edges for 96 nodes. Each node was associated with a 20-dimensional unit vector as pseudo-features. Additionally, we assigned a three-class categorical label to each node, categorizing them as 'strong,' 'medium,' or 'weak' influencers within their respective local communities. The label assignment was determined based on the nodes' degrees with cutoffs at 20 and 60. For example, a node with a degree of 30 would be classified as a 'medium influencer.'

To facilitate model training, we applied random masking to the training, validation, and test sets, ensuring equal proportions in each set. During the training process, we recorded the learned similarity scores $s_{ij}$ at the final layer for later use in generating the simplified network.

Figure 9 highlights a noticeable divergence between the top two histograms (representing the weight distributions in the original networks) and the bottom two histograms (depicting the weight distributions in the simplified networks). Specifically, a higher concentration of weights is observed at the extreme regions (with weights close to $0$ and $1$) in the simplified networks. This divergence underscores the impact of our simplification approach on the network's edge weight distribution.

## F ADDITIONAL BACKGROUND: CO-OCCURRENCE NETWORK OF METABOLIC GENES

The co-occurrence network of metabolic genes is a graph representation that illustrates the statistical associations and co-occurrence patterns among various metabolic genes within a biological system (Bello et al., 2020). This network emerges from computational analyses of extensive genomic data, with each node denoting metabolic genes linked to distinct biochemical functions, such as sugar production and TMAO (trimethylamine N-oxide) synthesis. These connections are quantified by the likelihood of two crucial functional genes co-occurring within the same species at a given time (Liu et al., 2022). Due to the complexity of the co-occurrence network of metabolic genes in different microorganism species, which arise from a large number of genes and connections, simplifying the network enables us to identify how key metabolic genes in microorganisms can be gathered into several interdependent modules. This is significantly important for revealing the mechanism of how genes work together within metabolic pathways, how they respond to different environmental conditions, and which genes may have essential roles in specific biological processes, highlighting the significance of the adaptation of microorganisms to changing environmental conditions (Levy & Borenstein, 2013).