# OpenReview forum: "A Unified View on Neural Message Passing with Opinion Dynamics for Social Networks"
_ICLR.cc/2024/Conference — Submitted to ICLR 2024_

### Official Review · Reviewer_gTLa · 2023-10-30

**Soundness:** 2 fair
**Presentation:** 3 good
**Contribution:** 2 fair
**Rating:** 5
**Confidence:** 4

**Summary:**

This paper introduces a method called ODNET that combines sociological concepts from social networks with message passing. It incorporates the concept of bounded confidence, dynamically adjusting the influence weight on target nodes based on their similarity between the target node and their neighbor nodes, which could better simulate the propagation and aggregation of information in graph structures. And the results shows that ODNET outperforms other graph neural network models in node classification tasks and decreases the over-smooth problem.  Furthermore, this method has also been successfully applied to various types of graphs in this paper, including heterophilic graphs, homophilic graphs, and hypergraphs. Lastly, through ODNET, it becomes possible to explain the internal information exchange within networks and the roles of genes in different metabolic pathways.

**Strengths:**

1.	This paper introduces a novel MP framework ODNET which employs the influence function with bounded confidence.
2.	The ODNET method outperforming other baseline GNN models including heterophilic graphs and homophilic graphs. And the ODNET is generalized into hypergraphs.
3.	The ODNET decreases the over-smooth problem in GNN models and explains the internal information exchange within networks and the roles of genes.

**Weaknesses:**

1.	The structure of the paper is not very clear. There are minor symbol errors in the text.
2.	The example of social network architecture simplification is discussing about the microbial comparison between the Mariana Trench and Mount Everest networks, which is not adequately explained that the connection to social networks is unclear.
3.	This method is not very innovational that it combines the mathematical models from sociology to update node representation in MP.

**Questions:**

Why the microbial comparison between the Mariana Trench and Mount Everest networks is put into social network architecture simplification? Is it possible to make it clear how to identify opinion leaders, bridge communicators and isolators through ODNET?

---

> ### Author Response · Authors · 2023-11-19
>
> **Weaknesses-1. Structure of the Paper**:
>
> After introducing the intuition of social network and opinion dynamics and their connection to graphs and MPNNs in the Introduction, we introduced the key concepts in GNNs (Section 2) and two types of models in opinion dynamics, as well as the connection of their formulation with respect to MPNNs and their properties with respect to the oversmoothing issue (Section 3). Based on the two models’ inherent characteristics, we then proposed ODNet with its discrete and continuous update schemes and generalized it for updating hypergraphs (Section 4). In section 5, we designed extensive experiments to validate ODNet from various perspectives. Section 6 then demonstrates the ability of ODNet to simplify social networks with practical biological examples. Section 7 and 8 review related works and concludes the work.
>
> **Weaknesses-2. The Connection of the Network Simplification Task and Social Networks in General**:
>
> Co-occurrence networks of metabolic genes characterize the likelihood that different genes occur simultaneously within the same microbial species. The nature of this co-occurrence indicates that the genes have potential interconnections with each other, thus forming a complete set of metabolic pathways to maintain normal microbial activity.
> The co-occurrence networks of metabolic genes can be considered a general form of social networks due to their structural and functional similarities (Nature Reviews Microbiology 2012, 10, 538-550;  BMC Genomics 2015, 16, S6; Microbiome 2020, 8, 82). Firstly, these networks reflect interactions and associations among genes, much like individuals forming relationships in a typical social network. In both cases, entities (genes or individuals) are connected and thus influence each other’s behavior or function. Secondly, typical social networks exhibit clusters of individuals with similar interests or affiliations, whereas metabolic gene networks showcase clusters of genes involved in specific metabolic pathways of functions. Lastly, both types of networks demonstrate resilience to perturbations, with redundancies in gene pathways ensuring functionality despite changes, akin to the way multiple connections in a social network can provide stability in the face of disruptions.
>
> Due to the large number of genes and their connections, it becomes particularly difficult to find the key genes and their connections to functional genes. Therefore, we need to develop tools that assist in exploring the nature of social interactions of genes. The tool should demonstrate ways to efficiently gather extensive information about these networks and present them visually in an effective manner. As for our work, by comparing the network structures after simplification, we can clearly observe whether microorganisms that survive in two distinct environments (Mount Everest vs. Mariana Trench) have similar ways of organizing their metabolic pathways.
>
>
> **Weaknesses-3. MPNN combines Sociology**:
>
> We have conducted a thorough search and found the only similar previous work that also mentioned opinion dynamics and MPNN in their paper is SINN (Okawa et. al., 2022, KDD). While both papers proposed MPNN update schemes inspired by opinion dynamics, our work is largely different from theirs from multiple perspectives.
>
> (1) We employed a potential term to avoid excessive attraction or repulsion from neighbors, so as to progressively update node representations. In this regard, our update rule prevents the energy from being continuously reduced, thus circumventing the oversmoothing issue.
>
> (2) We established the bounded confidence with scaling factors. In particular, we leveraged different scaling factors when aggregating the neighborhood information that balances the attractive or repulsive forces.
>
> (3) We further generalize our ODNet for hypergraph representation learning, which is a nontrivial extension from graph settings.

---

> ### Author Response · Authors · 2023-11-19
>
> **Questions-1. Comparison between MT&ME is an Architecture Simplification Task**:
>
> Due to the large number of genes and their complicated connections, analyzing metabolic gene networks is a difficult task. This difficulty makes it challenging to establish relationships between key genes and functional genes. Hence, there is a need to simplify the metabolic network.
>
> In this study, through the simplification of the original gene networks of the Mariana Trench and Mount Everest from our previous work (Microbiome, 10(1): 215, 2022), we discovered that in the MT network, the key gene nitrous oxide reductase (nos) bridged the carbon (Alkane, Aromatic, Complex sugar, D-AA, and L-sugar) and nitrogen metabolism (N) (Fig. 4c). Conversely, in the ME network, the key genes involved in sulfate reduction (sat and aprA) connected the carbon and sulfur metabolism (S) (Fig. 4d). This indicates that in different environments, the same functional genes couple with different key genes.
>
> It is notable that the analysis of genes and their connections in the metabolic network from different environments relies primarily on the simplified microbial social network architecture to facilitate meaningful comparison. While the genes are essentially the same in different networks, it requires the simplification algorithm to modify the network based solely on the initial connection of the graph.
>
>
> **Questions-2. Identification of Network Roles Through ODNet**:
>
> We drew Figure 4 with force-directed placement (Software - Practice and Experience, 21(11):1129-1164, 1991) based on the ODNet-learned edge weights. The algorithm allocates the nodes to concentrate nodes with connections while repulsing overly-closed nodes to avoid overlapping. Here, ODNet refines the edge weights of the network to influence the visualization from two perspectives. Firstly, it enhances connections among the genes in the same communities (i.e., genes exhibit similar functionality) and weakens or even cuts the connections between loosely connected genes. Consequently, genes within the same community will exhibit locally dense connections by ODNet and result in compact positioning in the visualization. Secondly, key genes that are connected to more genes with an above-average degree will be identified as the opinion leader or the bridge communicator and will be forced to be apart from other nodes in the visualization. Isolators, as implied by their names, are usually loosely connected to any other nodes with a small degree and small edge weights.

---

### Official Review · Reviewer_vhvr · 2023-11-04

**Soundness:** 1 poor
**Presentation:** 1 poor
**Contribution:** 2 fair
**Rating:** 3
**Confidence:** 2

**Summary:**

This paper proposes a new message passing scheme for Graph Neural Networks, inspired by the Hegselmann-Krause (HK) opinion dynamics model. It is claimed that the proposed model, ODNet, resolves the oversmoothing issue of GNNs. Experiments show that ODNet significantly outperforms selected GNN baselines on popular benchmarks such as Cora, Citeseer, Pubmed.

**Strengths:**

1. To study the connection between opinion dynamics and neural message passing scheme is an interesting idea.
2. Experiments show that ODNet has some edge compared to traditional GNNs, Figure 2 also provides an example on which ODNet significantly alleviates oversmoothing.

**Weaknesses:**

1. It still remains very unclear to me why opinion dynamics can be used to design Graph Neural Networks. Opinion dynamics describe some hypothesized laws that humans might apply when exchanging opinions with others in a social network. GNNs are a class of neural architectures for the sake of capturing graph signals and make accurate predictions. I can clearly see that both of them are passing messages on the graph structure, with each node aggregating information from neighboring nodes in each iteration. However, the high-level similarity in their working mechanism does not explain why GNNs should do message passing following the way how humans exchange opinions.

2. Eq. (6) (7) requires O(n^2) complexity to compute in each iteration of message passing, which abandons one of the most important characteristics of GNNs in leverage graph sparsity. Can the authors justify why this is a good choice, as well as the numerical comparison of ODNet's time complexity with other baselines?

3. The baselines used in experiments are outdated. Most GNN architectures are at least 3-5 years ago.

4. The readability of some key parts of the paper is extremely concerning. I find it very hard to understand, for example, the second paragraph on page 2 ("The opinion dynamics-inspired ...") and the paragraph of "Clustering and Oversmoothing in Herterophilious Dynamics" on page 4. Can the authors explain these two paragraphs in simpler language? For the former, why do microbial communities and metabolic genes suddenly appear in the context of social networks and opinion dynamics; for the latter, are the authors claiming that HK model does better on oversmoothing? I am extremely confused why so many things, "clustering", "oversmoothing", "heterophily", and "dirichlet energy", all show up together when none of them has been mentioned or eve hinted in the previous paragraph.

**Questions:**

See weaknesses.

**Details Of Ethics Concerns:**

I am not sure if this is something worthy of further investigation, and I understand that LLMs are allowed in writing. However, I feel that many places in this paper have too strong trace of the typical type of writing produced by LLMs, to an extent that both the idea's readability and originality is very concerning. For example, please see the second paragraph on page 2 ("The opinion dynamics-inspired ..."), and the paragraph of "Clustering and Oversmoothing in Herterophilious Dynamics" on page 4.

---

> ### Author Response · Authors · 2023-11-19
>
> **Weaknesses-1. Connection Between Opinion Dynamics and GNN**:
>
> Before further explaining our motivation for designing our message passing scheme with opinion dynamics, we would like to clarify some conceptual misunderstandings.
>
> Firstly, **GNNs are trained to update hidden representations for nodes, instead of making label predictions**. GNN layers extract high-level hidden representations for nodes to better describe the local environment. The prediction is made by extra read-out layers, such as MLPs for static graphs and LSTMs for spatial-temporal graphs. Also, there are cases where the label prediction is not complimentary. For instance, GAE is a classic network that optimizes the hidden representation that reconstructs the graph and graph connections.
>
> Also, **we do not design ODNet based on the belief that GNNs have to do message passing following the way humans exchange opinions**. Instead, we initially analyzed the similarity between opinion dynamics and existing MPNNs (e.g., GRAND) to demonstrate the inherent connections between these two different systems, i.e., the formulation of the two systems in describing the information exchange among agents (nodes). Based on this, we then leveraged the concept of bounded confidence and proposed a novel MPNN that (1) follows a simple yet effective construction, (2) is physically informed, and (3) alleviates the oversmoothing issue. We argued that a trivial combination of tools from different domains with fine-tuned ‘SOTA’ performance does not form an elegant research work. Instead, we aimed to propose a novel framework that is practically meaningful (e.g., can be applied for solving real-world scientific problems) with explainable (e.g., physically informed) constructions and certain theoretical guarantees (e.g., alleviating the oversmoothing issue).
>
>
> **Weaknesses-2. Additional Complexity in ODNet**:
>
> We recognize that leveraging graph sparsity is an important aspect of GNN. However, network expressivity is also important, and one should not sacrifice it only to pursue the sparse aggregation rule. In our case, we attached the potential term and the bounded confidence in eq. 6-7 to circumvent the oversmoothing issue. While it introduces extra complexity for the update rule in a single update step, it does not significantly increase the training time. For instance, on an A100 (single card), it costs ~50 seconds to complete the entire program for one iteration on Cora, whereas ~80 seconds is required for SINN (Okawa et. al., 2022, KDD).
>
>
> **Weaknesses-3. Baselines**:
>
> As the proposed ODNet follows a continuous update scheme, we selected representative discrete and continuous MPNN baseline methods for comparison, which were published between 2017-2022 with the majority released after 2020. As per the suggestion, we added two other baseline methods, SINN (2022, KDD) and ACMP (2023, ICLR). Both are continuous MPNN update schemes with neural ODE with public official PyTorch implementations. Due to the time limitation, we implemented the new baseline methods and reported their performance on the Citation networks. The reason is mainly that SINN requires additional time-consuming data processing for the `DataLoader`. We will update the rest results on node classification when they are available. Note that the existing methods do not apply to prediction tasks on hypergraphs as their pipeline requires additional non-trivial modifications.
>
> | Method  | Cora | CiteSeer | PubMed |
> |-------------|--------|-------------|-------------|
> | SINN      | 83.2±0.4 | 73.9±0.6 | 80.0±1.7 |
> | ACMP     | 84.9±0.6 | 75.0±1.0 | 78.9±1.0 |
> |ODNet      | 85.7±0.3 | 75.5±1.2 | 80.6±1.1 |

---

> > ### Author Response · Authors · 2023-11-19
> >
> > **Weaknesses-4. Readability**:
> >
> > (1) "The opinion dynamics-inspired ..." on page 2 means “the propagation scheme which is inspired by opinion dynamics…”. In this paragraph, we presented three examples of how interactions in opinions influence different types of social networks, including *gene co-occurrence networks* (for microbial communities, where in its graph representation, nodes are metabolic genes and edges connect interactive genes), *social network of politicians*, and *social network of scholars*. Here, a social network is roughly categorized based on its position on the opinion spectrum (visualized in Figure 1). In particular, microbial communities, together with political networks and scholar networks, are different types of social networks that we used as empirical examples to explain Figure 1 and the corresponding statements therein.
> >
> > (2) In the paragraph “Clustering and Oversmoothing in Heterophilious Dynamics” on page 4, we discuss the two phenomena, namely “clustering” and “oversmoothing”, on graphs or communities exhibiting heterophily. As two widely discussed concepts/phenomena, "oversmoothing" is introduced at the bottom of page 3, and “Dirichlet energy” is defined on top of page 4. "Heterophily", another widely-employed concept in social networks, is defined at the middle of page 4 as “agents tend to form stronger bonds with counterparts rather than with similar individuals”. "Clustering" literally means that the entities in a network tend to converge to one or several small communities. For instance, in machine learning, a representative unsupervised learning task is ‘clustering’, meaning dividing the unlabeled objects into several groups. Due to the page limit, we decided not to explain this basic concept in detail in the main text.
> >
> > (3) We did not claim that “the HK model does better on oversmoothing”, but “the HK model does better than the FD model in alleviating the oversmoothing issue”. This conclusion can easily be made by comparing sections 3.1 and 3.2. In particular, Section 3.1 details how the FD model achieves consensus with exponential decay of energy (i.e., oversmoothing in a graph). In Section 3.2, we referred to the construction of the HK model for a simple and effective potential solution to the oversmoothing issue in MPNN. To help understand the intuition and motivation behind this idea, we provided an explanation in the paragraph “Clustering and Oversmoothing in Heterophilious Dynamics”.
> >
> >
> > **Ethics Concerns**:
> >
> > We acknowledged in the submission that we used LLM ONLY for checking grammar. We would like to emphasize that all the content was drafted by ourselves, and any corrections suggested by LLM were carefully double-checked before applying them to the paper. In particular, the two example phrases/sentences mentioned by the reviewer were written by ourselves. The highlighted line on page 2 means “the propagation scheme which is inspired by opinion dynamics…”. In the paragraph “Clustering and Oversmoothing in Heterophilious Dynamics” on page 4, we connected the two phenomena, “clustering” (in social networks) and “oversmoothing” (in graph representation learning). We referenced the previous study on the clustering phenomenon in communities characterized by strong heterophily and provided an intuitive explanation of how the HK model alleviates the oversmoothing issue.

---

> > > ### Comment · Reviewer_vhvr · 2023-11-19
> > > **Thanks for the reponse**
> > >
> > > I have carefully read the author's response, and I appreciate the explanation and extra experiment. However, most of my concerns remain unresolved.
> > >
> > > 1. for W1, The paper's title is "A Unified View on Neural Message Passing with Opinion Dynamics for Social Networks". However, the work is essentially just borrowing the idea of HK model to alleviate oversmoothing. How could this qualify as a unified view?
> > >
> > > 2. For W2, the authors have been emphasizing this work as  a "practically meaningful" method. In this context, it is very hard to convince me that we should completely sacrifice "graph sparsity" as such to pursue expressiveness.
> > >
> > > 3. For W4, It is still extremely confusing why social networks should include gene co-occurrence networks.

---

> ### Author Response · Authors · 2023-11-20
>
> **W1**:
>
> We would like to clarify that this work is NOT ‘*essentially just borrowing the idea of HK model to alleviate oversmoothing*’, and we are not ambitiously aiming at bridging every single piece of MPNN and opinion dynamics (which is also impossible to be covered by a conference paper and there are not enough existing work to summarize at this stage). To this end, we will explain again the main workflow, in the hope of eliminating misunderstanding and confusion regarding the contribution and novelty of this research.
>
> (1) We started from the FD model and GRAND, two representative models in opinion dynamics and continuous MPNN, to show their intrinsic connections. Based on this, we provided a new explanation from sociology on why FD-like models fail to address the oversmoothing problem.
>
> (2) It naturally motivates us to refer to the ideas in sociology, e.g., bi-clustering, to design GNNs that alleviate the oversmoothing issue. In particular, we adopted the construction of bounded confidence and scaling factors from opinion dynamics to formulate our aggregation rule in its discrete and continuous form.
>
> (3) We then generalized ODNet to analyze hypergraphs, a less-explored type of graphs with typical features exhibited in social networks (e.g., strong local connectivity). Note that such a generalization is non-trivial, and our method achieves superior performance over existing baselines.
>
> (4) We made further theoretical analysis and discussions regarding the effectiveness of the collective behaviors in ODNet (Appendix A), which suggests potential future explorations in extending the idea in sociology and particle systems for designing and analyzing MPNNs.
>
>
> **W2**:
>
> We would like to clarify the two misunderstandings regarding the different understandings of a ‘practical meaningful’ method between us and the reviewer. Firstly, as we responded in the previous post, we believe our method is meaningful because it ‘*can be applied for solving real-world scientific problems*’. This statement has been validated in Section 6, where ODNet is able to simplify social networks. Secondly, we agree with the reviewer that scalability is one aspect that empowers certain types of GNNs in some applications and it is desired to have a computationally efficient tool to interpret large-volume data **if the model performance does not hurt significantly and the scalability is the primary concern of the application**. We argue that:
>
> (1) expressivity is more important than graph sparsity especially when the trained graph is not crazily large, e.g., over billions of nodes.
>
> (2) we do not completely sacrifice graph sparsity, as believed by the reviewer. As a matter of fact, ODNet simplifies the graph connection to delete unnecessary connections. For instance, in the gene co-occurrence network simplification task, the original ME network includes 2517 edges. After simplification, the remaining number of edges is 260 (with a cutoff of 0.05).
>
> (3) In the context of node representation learning and social network simplification, large-volume graphs do not dominate real-world applications. The majority of the existing graph benchmark datasets contain fewer than a million nodes. For the simplification task, as it aims at understanding and explaining the relationship within the network, it is unrealistic to imagine the analysis being done on more than hundreds of thousands of nodes. Consequently, scalability is not the primary concern in our study.

---

> > ### Author Response · Authors · 2023-11-20
> >
> > **W4**:
> >
> > Co-occurrence networks of metabolic genes characterize the likelihood that different genes occur simultaneously within the same microbial species. The nature of this co-occurrence indicates that the genes have potential interconnections with each other, thus forming a complete set of metabolic pathways to maintain normal microbial activity.
> > The co-occurrence networks of metabolic genes can be considered a general form of social networks due to their structural and functional similarities (Nature Reviews Microbiology 2012, 10, 538-550;  BMC Genomics 2015, 16, S6; Microbiome 2020, 8, 82). Firstly, these networks reflect interactions and associations among genes, much like individuals forming relationships in a typical social network. In both cases, entities (genes or individuals) are connected and thus influence each other’s behavior or function. Secondly, typical social networks exhibit clusters of individuals with similar interests or affiliations, whereas metabolic gene networks showcase clusters of genes involved in specific metabolic pathways of functions. Lastly, both types of networks demonstrate resilience to perturbations, with redundancies in gene pathways ensuring functionality despite changes, akin to the way multiple connections in a social network can provide stability in the face of disruptions.
> >
> > Due to the large number of genes and their connections, it becomes particularly difficult to find the key genes and their connections to functional genes. Therefore, we need to develop tools that assist in exploring the nature of social interactions of genes. The tool should demonstrate ways to efficiently gather extensive information about these networks and present them visually in an effective manner. As for our work, by comparing the network structures after simplification, we can clearly observe whether microorganisms that survive in two distinct environments (Mount Everest vs. Mariana Trench) have similar ways of organizing their metabolic pathways.

---

### Official Review · Reviewer_Qd4r · 2023-11-09

**Soundness:** 3 good
**Presentation:** 2 fair
**Contribution:** 2 fair
**Rating:** 5
**Confidence:** 4

**Summary:**

This paper proposes to integrate the dynamics of opinion defined in some opinion dynamics models such as Degroot model and the Hegeselmann and Krausse model to propose new aggregation equations for GNNs. The authors then combine these dynamics with the Neural ODE paper to train the parameters of the resulting model. The authors report improvement over node prediction tasks, and other tasks over baseline GNNs.

A key contribution of the paper is the design of the phi function which can incorporate homophily and heterophily. However they also introduce two new parameters. While their significance has been explained, how to set those parameters is still not intuitively clear. Specifically, on an unknown graph we may not have any idea about the nature of interactions that led to the graph.

Another contribution seems to be the integration of the whole model into the neural ODE framework for learning. However, the authors assume familiarity of the reader with this framework. It is very difficult judge the added complexity due to this addition. The authors also do not report training times and depth to which these networks and baseline models can be trained. Also, what about other GNN tasks e.g. link-prediction.

The author should also compare and contrast the role of other literature on learning of opinion dynamics models using neural networks:
Okawa, Maya, and Tomoharu Iwata. "Predicting opinion dynamics via sociologically-informed neural networks." In Proceedings of the 28th ACM SIGKDD Conference on Knowledge Discovery and Data Mining, pp. 1306-1316. 2022.

**Strengths:**

The design of aggregation function phi.
Experimental results on node prediction.

**Weaknesses:**

Limited scope of experimentation. Only node classification problem is addressed. Also the reason for good results is not sufficiently explored.
Missing literature review. A large class of methods in opinion dynamics has not been referred. Also, it is not clear why some of the other referred collective dynamics references are not effective.
Also, the overall contribution seems limited.

**Questions:**

None.

---

> ### Author Response · Authors · 2023-11-19
> **response to Reviewer Qd4r**
>
> **Training times and depth**:
>
> As we did not use the term ‘training depth’ within the paper, we are uncertain if it refers to the number of network layers in the review. If this is the case: we do not fix the number of layers for any datasets. Instead, it adapts to the ODE solver. In terms of training time, for your reference, on Cora, the entire program takes ~50 seconds for ODNet for one repetition, whereas for SINN it takes ~80 seconds to complete one repetition (excluding the additional data processing time). For other baseline methods mentioned in Section 5.1-5.2 (and Appendix B), the benchmarks are directly loaded from PyG or follow pre-processing from previous work, and the results are sourced from previous studies.
>
>
> **Comparison with Okawa *et al*.’s work**:
>
> We have provided a detailed comparison with SINN (Okawa et. al., 2022) in the revised version. In summary, both papers proposed MPNN update schemes inspired by opinion dynamics. However, our work is different from theirs from three perspectives.
>
> (1) We employed a potential term to avoid excessive attraction or repulsion from neighbors, so as to progressively update node representations. In this regard, our update rule prevents the energy from being continuously reduced, thus circumventing the oversmoothing issue.
>
> (2) We established the bounded confidence with scaling factors. In particular, we leveraged different scaling factors when aggregating the neighborhood information that balances the attractive or repulsive forces.
>
> (3) We further applied our ODNet for hypergraph representation learning, which is a nontrivial extension from graph settings.
> Empirically, we implemented SINN for node classification tasks based on its official implementation at (https://github.com/mayaokawa/opinion_dynamics). As it requires additional time-consuming data processing for the `DataLoader`, we were only able to perform the program on Citation networks within the limited time constraint. We will update the rest results on node classification when they are available. Also, it is notable that the current version of SINN does not apply to prediction tasks on hypergraphs, as its pipeline requires additional sophisticated modification.
>
> | Method  | Cora | CiteSeer | PubMed |
> |-------------|--------|-------------|-------------|
> | SINN      | 83.2±0.4 | 73.9±0.6 | 80.0±1.7 |
> |ODNet      | 85.7±0.3 | 75.5±1.2 | 80.6±1.1 |

---

> > ### Author Response · Authors · 2023-11-19
> >
> > **Scope of Experimentation**:
> >
> > We would like to clarify that our experimental analysis is carefully designed with **extensive benchmarks of different characteristics** and **a comprehensive analysis from different perspectives** to suit the overall purpose of this research.
> >
> > (1) First, we covered a wide range of social networks in the form of homophilic graphs, heterophilic graphs, and hypergraphs. The first two types of graphs demonstrate how the position of a graph on the opinion spectrum influences its update rule, so as to impact the empirical performance on the prediction results. The last task on hypergraphs, with typical features exhibited in social networks (i.e., strong local connectivity) requires a non-trivial generalization from GNN update rules, yet our method achieves superior performance over existing baselines.
> >
> > (2) We then conducted an extensive investigation of the proposed ODNet regarding its features and settings. In particular, we examined the decay of Dirichlet energy of our method and compared it to other MPNN methods (e.g., GCN, GAT, and GRAND) to show that our method can indeed circumvent the oversmoothing issue as expected. We also discussed the influence of hyperparameters, such as influence functions, scaling factors, and ODE solvers, for prediction performance. Furthermore, we visualized the embedding dynamics of the model to intuitively show the internal changes during the training process.
> >
> > (3) Our method is not only theoretically sound and surpasses baseline methods in standard prediction tasks on benchmark datasets, but it also well-addresses practical tasks in social networks in terms of network simplification. As a working example, we provided a thorough analysis of gene co-occurrence networks, where we simplified two networks of the same group of genes from different external environments based on their complete connection initialization. In summary, we designed the experiments to demonstrate our proposed ODNet is practically meaningful (e.g., can be applied for solving real-world scientific problems) with explainable (e.g., physically informed) constructions and certain theoretical guarantees (e.g., alleviating oversmoothing).
> >
> > In terms of link prediction tasks, we argued that it is not the main focus of this work and it does not fit the update target of our model. A key aspect of ODNet is to reallocate the edge weights of the current network so as to concentrate agents around a few representative individuals. In social networks, this phenomenon is well-known as “scale-free”. While it fits a lot of social networks and forms an efficient message-exchanging route through reformulating the network, the ‘modification’ nature of this update rule does not fit the nature of link prediction tasks, which aim at ‘recovering potential connections’ no matter if they are strongly or loosely linked to each other.
> >
> > **Literature Review**:
> >
> > To the best of our knowledge, SINN is the only research that combines opinion dynamics and MPNN, for which the difference with our model has been explained in the previous response. We have included this discussion in the revised version and would be very grateful if the reviewer could help us identify other papers from “a large class of methods in opinion dynamics” that we missed for MPNN designs.
> >
> >
> > **The Effectiveness of Other Collective Dynamics**:
> >
> > We would like to clarify that we did not claim that the HK model is the only choice for effective marriage with MPNNs, and this study is not aimed at exhaustively comprehending all possible choices in collective dynamics to construct their respective MPNN update rules. Instead, we only analyzed the collective behaviors in HK-driven models in effectively alleviating the oversmoothing issue in the main text and suggested that “When devising MP aggregation rules, employing a comprehensive framework for microscopic models based on collective behaviors within a complete interaction system proves effective in mitigating the oversmoothing issue.” in Appendix A.

---

> ### Author Response · Authors · 2023-11-19
>
> **Overall Contribution**:
>
> We hope the reviewer could agree with the overall contribution after our clarification. In summary:
>
> (1) Although SINN (which is the only existing work that expands opinion dynamics to MPNNs to the best of our knowledge) designed MPNNs from the perspective of sociology, their method is significantly different from ours from methodology design, model functionality, and applications.
>
> (2) We designed our experiments with extensive investigations to well-suited our main claims from different perspectives, which goes beyond simply node classification tasks on graph benchmark datasets.
>
> (3) On top of baseline comparisons, we conducted further investigations to analyze why our model surpasses other baseline methods and how it achieves the target of network simplification with an empirical example in biology. We tried our best to comprehend theoretical and empirical analysis to support the novelty of the proposed method, demonstrating that it is practically meaningful with explainable constructions and certain theoretical guarantees.

---

### Meta-Review · Area_Chair_wFwr · 2023-11-27

**Metareview:**

All the reviewers were negative about the paper and the rebuttal period did not fully clear out the concerns they had.

**Justification For Why Not Higher Score:**

All the reviewers were negative about the paper.

**Justification For Why Not Lower Score:**

N/A

---

### Decision · Program_Chairs · 2024-01-16

Reject